# Mini-Batch Consistent Slot Set Encoder for Scalable Set Encoding

**Andreis Bruno**[1], **Jeffrey Ryan Willette**[1], **Juho Lee**[1,2], **Sung Ju Hwang**[1,2]

KAIST [1], South Korea
AITRICS [2], South Korea
`{andries, jwillette, juholee, sjhwang82}@kaist.ac.kr`

## Abstract

Most existing set encoding algorithms operate under the implicit assumption that all the set elements are accessible, and that there are ample computational and memory resources to load the set into memory during training and inference. However, both assumptions fail when the set is excessively large such that it is impossible to load all set elements into memory, or when data arrives in a stream. To tackle such practical challenges in large-scale set encoding, the general set-function constraints of permutation invariance and equivariance are not sufficient. We introduce a new property termed Mini-Batch Consistency (MBC) that is required for large scale mini-batch set encoding. Additionally, we present a scalable and efficient attention-based set encoding mechanism that is amenable to mini-batch processing of sets, and capable of updating set representations as data arrives. The proposed method adheres to the required symmetries of invariance and equivariance as well as maintaining MBC for any partition of the input set. We perform extensive experiments and show that our method is computationally efficient and results in rich set encoding representations for set-structured data.

## 1   Introduction

Recent interest in neural network architectures that operate on sets [14, 6] has garnered momentum given that many problems in machine learning can be reformulated as learning functions on sets. Problems such as point cloud classification [13], image reconstruction [3, 4, 7], classification, set prediction [8], and set extension can all be cast in this framework of learning functions over sets. Given that sets have no explicit structure on the set elements, such functions are required to conform to symmetric properties such as permutation invariance or equivariance for consistent processing.

A defining property of many practical functions over sets involves an *encoding* of the input set to a vector representation, the *set encoding*. This set encoding is then used for downstream tasks such as reconstruction or classification. In DeepSets [14], a sum-decomposable family of functions is derived for a class of neural network architectures that encode a given set to such a representation. However, the simplicity of the functions derived in DeepSets makes it ineffective for modeling pairwise interactions between the elements of the sets. Set Transformers [6] remedy this by using Transformers [12] to model higher order interactions resulting in rich and expressive set representations.

In all these works, there is an implicit assumption in the experimental setup that the cardinality of the set is manageable, and that there are ample computational resources available for processing all the elements during the set encoding process. However in real-world applications such as large scale point cloud classification and data intensive applications in particle physics, the set size can be extremely large. In such cases, even if one has access to a set encoding function that is linear w.r.t. the number of elements in the set, it is still impossible to encode such sets in a single batch since we may not be able to load the full set into memory. Current encoding methods deal with this issue by sampling

35th Conference on Neural Information Processing Systems (NeurIPS 2021).

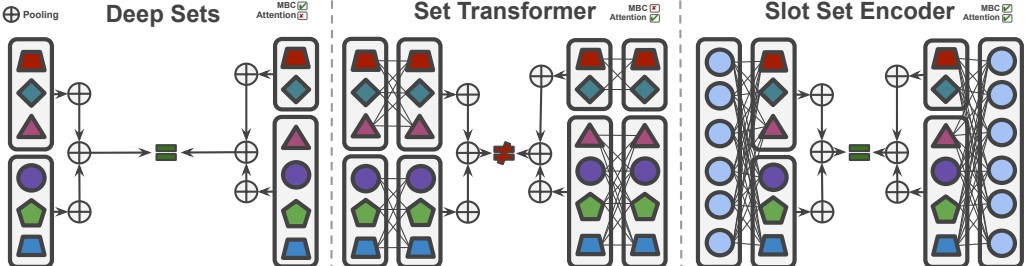

Figure 1: Mini-Batch Consistent Set Encoding. Each figure shows the processing of **(left)** mini-batch encoding of $\{s_1, s_2\} \in S$, and **(right)** $\{s'_1, s'_2\} \in S$. **Deep Sets:** can be made to encode batches consistently but cannot model pairwise interactions. **Set Transformer:** self-attention prevents mini-batch consistent set encoding. **Slot Set Encoder:** Attention w.r.t. parameterized slots allows for both attention and mini-batch consistency.

subsets of the full set (as a data preprocessing step) and encoding them to a representative vector, as done in the ModelNet40 [13] experiments in [14, 6]. However, this results in the loss of information about the full set, which is undesirable when huge amounts of monetary resources are invested in obtaining data such as in the Large Hadron Collider (LHC) experiments, and hence is imperative that the data is exploited for maximum performance on the given task. As such, it is necessary to have a set encoder that can take advantage of the full data, in a computationally efficient way.

To this end, when the set is too large to fit in the memory, or given as a stream of data, we propose to *iteratively* encode it over multiple rounds, by randomly partitioning it, encoding each random subset (mini-batch) with the set encoder at each round, and aggregating all the encodings. However, naive mini-batch set encoding could compromise essential properties of the set encoder such as permutation invariance and equivariance. To use a concrete example, in Figure 2 we show the performance of Set Transformer (in Negative Log-Likelihood) on an image reconstruction task. By processing the full set (in this case concatenations of pixel and coordinate values), the model performs as expected (*SetTransformer* in Figure 2). However when we instead partition the set elements into mini-batches, independently encode each batch and aggregate them to obtain a single set encoding, we observe performance degradation in the model as shown in Figure 2 (*SetTransformer(MiniBatch)* in Figure 2).

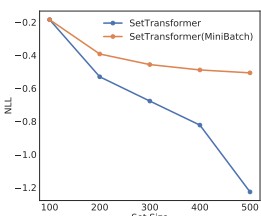

Figure 2: Performance degradation (CelebA image reconstruction) when evaluating Set Transformer [6] in a minibatch vs. non-minibatch setting.

This degradation stems from the attention module in Set Transformer where mini-batch encoding and aggregation results in mini-batch inconsistent set encoding.

In this work, we identify and formalize a key property, *Mini-Batch Consistency* (MBC), that is necessary for mini-batch encoding of sets. In what follows, we present the necessary constraints required for MBC and propose a *slot* based attentive set encoder, the *Slot Set Encoder* (SSE) which is MBC. SSE largely alleviates the performance degradation in mini-batch encoding of large sets. We depict MBC and non-MBC set encoders in Figure 1.

Our contributions in this work are as follows:

- We identify and formalize a key property of set encoding functions called Mini-Batch Consistency (MBC), which requires that mini-batch encoding of a set is provably equal to the encoding of the full set (in Section 2.2).

- We present the first attention based set encoder that satisfies MBC. Our method can efficiently scale to arbitrary set sizes by removing the dependence of self-attention in the transformer through the use of *slots* which frees the transformer from requiring to process the full set in a single pass (in Section 2.3).

- We perform extensive experiments on various tasks such as image reconstruction, point cloud classification and dataset encoding where we demonstrate that SSE significantly outperforms the relevant baselines (Section 3).

## 2 Mini-Batch Consistent Set Encoding with Slot Set Encoders

### 2.1 Preliminaries

Suppose we are given a dataset of *sets* $\mathcal{D} = \{X_i, \ldots, X_N\}$ where $|\mathcal{D}| = N$ and each $X_i \in \mathbb{R}^{n_i \times d}$ where $n_i$ is the number of elements in $X_i$ and each element $x_i^{(j)}$ (that is, the $j$th element of $X_i$) is represented by a $d$ dimensional tensor. For each $X_i$, the ordering of the $n_i$ individual elements is considered arbitrary and could be any permutation of the indices. We further assume that both $n_i$ and $d$ can be large enough such that processing an instance $X_i$ is prohibitive both in terms of memory and computation. As an example, the processing task could be encoding each element $X_i$ to a representative vector to be used for downstream tasks such as set reconstruction, set classification or set prediction. Current set encoding methods such as Zaheer et al. [14] and Lee et al. [6] deal with such large sets by sampling a smaller subset of the original set as a preprocessing step. In essence, they approximate the full set with a random subset which can be undesirable when we want to exploit all the data at our disposal for maximum performance. In this section, we present a method for encoding arbitrarily sized set data via a mini-batch encoding scheme. Our method can then *iteratively* encode subsets of $X_i$ and aggregate the subset-encodings to obtain the full set representation. Our method is *Mini-Batch Consistent*, invariant to the order of the subsets, and invariant to permutations on the set elements. For clarity, when we say a set $X_i$ is mini-batch processed, we mean that a random partition function can be applied to $X_i$ to obtain proper subsets of $X_i$. The subsets, which we call mini-batches, are then processed independently and aggregated to obtain a representation for $X_i$.

### 2.2 Mini-Batch Consistent Set Encoding

We consider a single element $X_i \in \mathcal{D}$ with $n_i$ elements each with $d$ dimensions. We truncate notation and write $X_i$, $n_i$ as $X$ and $n$ respectively since we present our algorithm for a single sample here. Our ultimate goal is to encode the set $X$ to a representation $Z \in \mathbb{R}^{d'}$ where $d'$ is the dimension of the encoding vector. We assume that both $d$ and $n$ are large such that it is not feasible to process $X$ as a whole. To get around this computational problem, we propose to perform mini-batch encoding of $X$ where the mini-batch samples are with respect to the number of elements $n$. Specifically, we partition $X$ such that $X$ can be written as a finite union $X = X_1 \cup X_2 \cup \ldots \cup X_p$ and each $X_i$ in this partition has $n_i = |X_i|$ elements and $p$ is the total number of partitions. We further assume that $X$ is partitioned such that we can efficiently process a given *mini-batch* $X_i$ in the partition. Each partition $X_i$ is encoded with a set encoding function $f(X_i)$ to obtain $Z_i$. We then define the set encoding problem as follows:

$$Z = g\big(f(X_1), f(X_2), \ldots, f(X_p)\big) \tag{1}$$

where $g(\cdot)$ is an *aggregation* function that takes individual $Z_i$'s and combines them to obtain $Z$. We further require that the functions $f$ and $g$ are *Mini-Batch Consistent*.

**Property 2.1** (Mini-Batch Consistency). *Let $X \in \mathbb{R}^{n \times d}$ be partitioned such that $X = X_1 \cup X_2 \cup \ldots \cup X_p$ and $f : \mathbb{R}^{n_i \times d} \mapsto \mathbb{R}^{d'}$ be a set encoding function such that $f(X) = Z$. Given an aggregation function $g : \{Z_j \in \mathbb{R}^{d'}\}_{j=1}^p \mapsto \mathbb{R}^{d'}$, $g$ and $f$ are Mini-Batch Consistent if and only if*

$$g\big(f(X_1), \ldots, f(X_p)\big) = f(X)$$

Property 2.1 ensures that no information is lost by encoding the full set in independent mini-batches, and that the aggregation function $g$ guarantees the same output as encoding the full set in a single batch. Additionally, Property 2.1 is general and places no restriction on the form of the functions $f$ and $g$ except that when used together, MBC must be satisfied. In what follows, we define an attention-based set encoding function $f$ together with aggregation functions $g$ that satisfy Property 2.1.

### 2.3 Slot Set Encoder

In this section, we provide a formulation for an attention based set encoding function $f$ in Equation 1 which utilizes slots [8]. Given an input set $X \in \mathbb{R}^{n \times d}$, we begin by sampling a random initialization for $K$ slots $S \in \mathbb{R}^{K \times h}$ where $h$ is the dimension of each slot. Specifically,

$$S \sim \mathcal{N}(\mu, \text{diag}(\sigma)) \in \mathbb{R}^{K \times h}, \tag{2}$$

where $\mu \in \mathbb{R}^{1 \times h}$ and $\sigma \in \mathbb{R}^{1 \times h}$ are learnable parameters. Optionally, instead of sampling random initialization for the $K$ slots, we can instead designate deterministic learnable parameters $S_\chi \in \mathbb{R}^{K \times h}$. An advantage of random initialization over learnable parameters is that at test time, we can increase or decrease $K$ for the randomly initialized model. In the ablation studies in section 3.1.1, we explore the performance gains that result from the choice of slot initialization. The key design choice in the slot set encoder is that unlike attention based set encoding methods like Lee et al. [6], we compute attention over $S$ (no self-attention) instead of computing it over the $n$ elements of $X$ (self-attention). This allows us to remove the dependence of the set encoding function on $n$, alleviating the requirement of processing the entire set at once, and allowing for MBC encoding of partitions of $X$ since the *same* slots are used to encode *all* mini-batches. Specifically, we compute dot product attention between $S$ and $X$ according to the following formulation:

$$\text{attn}_{i,j} := \sigma(M_{i,j}) \text{ where } M := \frac{1}{\sqrt{\hat{d}}} k(X) \cdot q(S)^T \in \mathbb{R}^{n \times K} \tag{3}$$

where $k$ and $q$ are linear projections of $X$ and $S$ to a common dimension $\hat{d}$ and $\sigma$ is the sigmoid activation function. There is a key difference in our attention formulation in that we use a sigmoid instead of a softmax to construct the attention matrix. Using a softmax, even when computed over S, breaks MBC as it requires a batch dependent normalization term which will vary across different partitions of $X$. By using the sigmoid function, we are able to free our model of this constraint and satisfy MBC. Finally, we weight the inputs based on the computed attention matrix to obtain the final slot set encoding $\hat{S}$ for $X$:

$$\hat{S} := W^T \cdot v(X) \in \mathbb{R}^{K \times \hat{d}} \text{ where } W_{i,j} := \frac{\text{attn}_{i,j}}{\sum_{l=1}^{K} \text{attn}_{i,l}} \tag{4}$$

where $v$ is also a linear projection applied to $X$ and $W \in \mathbb{R}^{n \times K}$ are the weights computed over slots instead of elements. The normalization constant in $W$ occurs over the slot dimension $K$, and only contains a linear combination of a single $n_i \in X$ and slots $S$ and therefore has no dependence on other elements within the partition $X_i$. The Slot Set Encoder is fully described in Algorithm 1.

The Slot Set Encoder in Algorithm 1 is functionally composable over partitions of the input set $X$. More concretely, for *a given* slot initialization and *any* partition of $X$,

$$f(X) = g(f(X_1), f(X_2), \ldots, f(X_p)) \tag{5}$$

Where the aggregation function $g$ is chosen to satisfy Property 2.1. This is convenient since it allows us to define $g$ from the following set of operators: $g \in \{\texttt{mean, sum, max, min}\}$. We note that to satisfy MBC, we have chosen $g$ to be an associative function but it is entirely possible to formulate a non-associative $g$ and corresponding $f$ that satisfies MBC and we leave that for future work.

**Proposition 2.1.** *For a given input set $X \in \mathbb{R}^{n \times d}$ and slot initialization $S \in \mathbb{R}^{K \times d}$, the functions $f$ and $g$ as defined in Algorithm 1 are Mini-Batch Consistent for any partition of $X$ and hence satisfy Property 2.1.*

**Proposition 2.2.** *Let $X \in \mathbb{R}^{n \times d}$ and $S \in \mathbb{R}^{K \times d}$ be an input set and slot initialization respectively. Additionally, let $\texttt{SSE}(X, S)$ be the output of Algorithm 1, and $\pi_X \in \mathbb{R}^{n \times n}$ and $\pi_S \in \mathbb{R}^{K \times K}$ be arbitrary permutation matrices. Then,*

$$\texttt{SSE}(\pi_X \cdot X, \pi_S \cdot S) = \pi_S \cdot \texttt{SSE}(X, S)$$

Proofs of Proposition 2.1 & 2.2 involve showing that each component in Algorithm 1 are both MBC, and permutation invariant or equivariant. Details can be found in the Appendix with discussions on why the baselines fail or satisfy MBC.

## 2.4   Hierarchical Slot Set Encoder

The Slot Set Encoder can encode any given set $X \in \mathbb{R}^{n \times d}$ to a $\hat{d}$ dimensional vector representation. In the mini-batch setting, all partitions are *independently* encoded and aggregated using the aggregation function $g$. In many practical applications, it is useful to model pairwise interactions between the elements in the given set since not all elements contribute equally to the set representation. Indeed this is the key observation of Lee et al. [6] which models such pairwise and higher order interactions between set elements and obtain significant performance gains over Zaheer et al. [14] which assumes

that all elements contribute equally to the set encoding. In the Slot Set Encoder, we would like to be able to model such pairwise and higher order interactions. However to satisfy Mini-Batch Consistency, we cannot model interactions between set elements given that the Slot Set Encoder removes all dependencies on $n$. Hence, we propose to model interactions among *slots* via a stack of Slot Set Encoders. Specifically, instead of using a Slot Set Encoder with $K = 1$, we stack $T$ Slot Set Encoders each with $K_i$ slots and set $K_T = 1$ for the final encoder. Concretely, if we let SSE be an instance of the Slot Set Encoder defined in Algorithm 1, then we can define a series of such functions, $\text{SSE}_1, \ldots, \text{SSE}_T$. Then for a set $X$, a composition of these Slot Set Encoders is a valid set encoding function that satisfies all the requirements outlined in Section 2.3, such that:

$$f(X) = \text{SSE}_T(\ldots \text{SSE}_2(\text{SSE}_1(X))) \tag{6}$$

This gives a *hierarchy* of Slot Set Encoders capable of modeling pairwise and higher order interactions between slots. The Hierarchical Slot Set Encoder is analogous to stacking multiple Induced Set Attention blocks followed by a final Pooling MultiHead Attention used in Set Transformer [6]. The major difference is that our model still remains MBC, a property violated by Set Transformer.

## 2.5 Approximate Mini-Batch Training of Mini-Batch Consistent Set Encoders

Set encoding mechanisms such as Zaheer et al. [14] and Lee et al. [6] require that gradient steps are taken with respect to the full set. However, in the Mini-Batch Consistent setting described so far, this is not feasible given large set sizes or constraints on computational resources. A solution to this problem is to train on partitions of sets sampled at each iteration of the optimization process. We verify this approach empirically and show that the Slot Set Encoder presented so far can be trained on partitions of sets and still generalize to the full set at test time. In these experiments (Figure 3b), we train set encoders on subsets of sets sampled at each optimization iteration and perform inference on the full set. Specifically, at some iteration $t$, we sample a mini-batch of

---

**Algorithm 1** Slot Set Encoder.
Partitioned input $X = \{X_1, \ldots, X_p\}$
Initialized slots $S \in \mathbb{R}^{K \times h}$
aggregation function $g$.

---

1: **Input:** $X = \{X_1, X_2, \ldots, X_p\}, S \in \mathbb{R}^{K \times h}, g$
2: **Output:** $\hat{S} \in \mathbb{R}^{K \times d}$
3: **Initialize** $\hat{S}$
4: $S = \text{LayerNorm}(S)$
5: **for** $i = 1$ **to** $p$ **do**
6:     Compute $\text{attn}_i(X_i, S)$ using Equation 3
7:     Compute $\hat{S}_i(X_i, \text{attn}_i)$ using Equation 4
8:     $\hat{S} = g(\hat{S}, \hat{S}_i)$
9: **end for**
10: **return** $\hat{S}$

---

size $B \times \tilde{n} \times d$, where $B$ is the batch size and $\tilde{n}$ is the cardinality of a partition of a set with $n$ elements ($\tilde{n} < n$). For this empirical analysis, we use the CelebA dataset [7] where the pixels in an image forms a set and the task is to reconstruct the image using the set encoding of a few pixels given as context points.

## 3 Experiments

We evaluate our model on the ImageNet [2], CelebA [7], MNIST [5] and ModelNet40 [13] datasets. Details on these datasets can be found in the Appendix. In these experiments, all sets arrive in a streaming setting in a mini-batch fashion where we only get subsets (batches) of the full set at an instance. Once the mini-batch is encoded, we only have access to the set encoding vector (we do not keep the original set elements since it requires further storage) which must be updated as more batches of the set arrive. We provide further implementation details in the Appendix.

### 3.1 Image Reconstruction

In this task, we are given a set of pixels (each 3 dimensional) with their corresponding coordinate points (each 2 dimensional), termed context values, and the task is to reconstruct the full image given the coordinates of the pixels we wish to predict using only the context points. The context points are 5 dimensional vectors formed by concatenating the pixels and their corresponding coordinates. We use Conditional Neural Processes [3], CNP, for this task. CNP has a set encoding layer for compressing the context points into a single vector representation that is then passed to a decoder for constructing the full image. Here, our goal is to demonstrate that the Slot Set Encoder is a proper set encoding

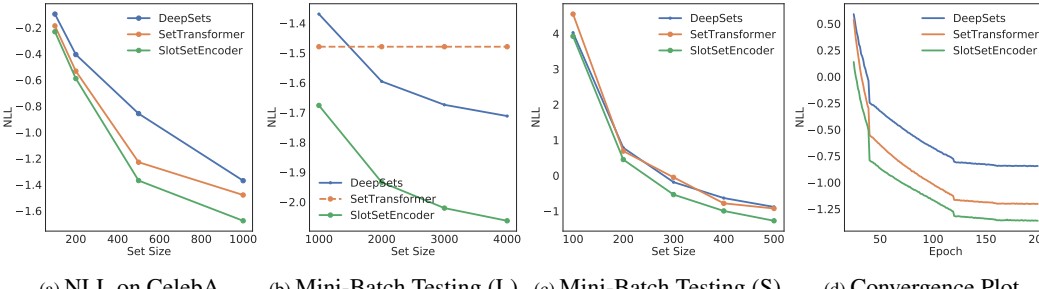

| (a) NLL on CelebA | (b) Mini-Batch Testing (L) | (c) Mini-Batch Testing (S) | (d) Convergence Plot |

Figure 3: **(a)** NLL for different set sizes (full sets, no mini-batch processing is used here) on the image reconstruction task. **(b)** Mini-Batch Testing of the reconstruction model (1000 set sized model in Figure 3a) trained on 1000 pixels and evaluated on large (L) set sizes of 1000-4000 pixels processing mini-batches of size 1000 at a time. **(c)** Mini-Batch Testing of reconstruction model(same as in Figure 3b) on smaller (S) set sizes of 100-500 pixels processing mini-batches of size 100 at a time. **(d)** Convergence plots for the considered Set Encoders on the image reconstruction task with the Slot Set Encoder trained according to Section 2.5.

function capable of learning rich set representations. We sample 200, 400, and 500 context points for images of size $32 \times 32$ and 1000 context points for images of size $64 \times 64$. As baselines, we use the mean set encoding mechanism of Zaheer et al. [14] and the Pooling MultiHead Attention (PMA) Blocks of Lee et al. [6]. We compare this with a variant of the Slot Set Encoder with random slot initialization. All the methods encode the given set to a 64 dimensional representative vector and are trained for 200 epochs. Additionally, for fair comparison, we only use a single Slot Set Encoder in our models as well as a single PMA block for Set Transformers.

The results for these experiments are presented in Figure 3a where it can be seen that the Slot Set Encoder model learns richer set representations resulting in lower negative log-likelihood when trained on full sets. Additionally in Figure 3b we demonstrate the scalability of the Slot Set Encoder where models initially trained on 1000 context points (in Figure 3a) are used to encode larger sets of context points (2000 - 4000) at test time. Since Set Transformer is not Mini-Batch Consistent, it cannot make use of the additional context points as it violates Property 2.1 hence in the constraints of the experiments, we use a 1000 context points for Set Transformer. DeepSets and Slot Set Encoders can utilize the additional data but the representation obtained by the Slot Set Encoder is richer in its representation power resulting in significant performance gains at test time as well as better generalization to larger set sizes than was used during training. In Section 3.1.1 we further discuss Figure 3b in the context of constrained resources and streaming data.

### 3.1.1 Ablation

The Slot Set Encoder presented so far has various components and evaluation modes that can have an impact on the richness of the resulting set representation. In this subsection, we perform extensive ablation studies to explore how these various components contribute to the performance of the encoder.

**Constrained Resources and Streaming Data** In order to demonstrate the utility of the models that satisfy Property 2.1, we simulate cases where the computational resources are limited in terms of memory. Specifically, we assume that our computational budget allows us to only compute the set representation for 1000 context points at a time. Under these constraints, the Set Transformer can only be used to encode 1000 context points since the Softmax attention layer requires all the set elements to be in memory. For MBC DeepSets, and Slot Set Encoder, we can use mini-batches of of size 1000 and iteratively update the set representation until the full set has been encoded. Additionally, this setting is akin to the case of streaming data where the set elements arrive or are obtained at irregular intervals and hence the set encoding representation must be incrementally updated.

We present the results of this experiment for an input image with 4096 pixels with 1000 context points randomly selected and encoded at each time step. From Figure 3b it can be seen that for both DeepSets and the Slot Set Encoder, the more batches we encode, the better the performance. However, the Slot Set Encoder significantly outperforms DeepSets due to it's ability to model interactions between the set elements and slots. Additionally, it can be seen that the MBC models get better as

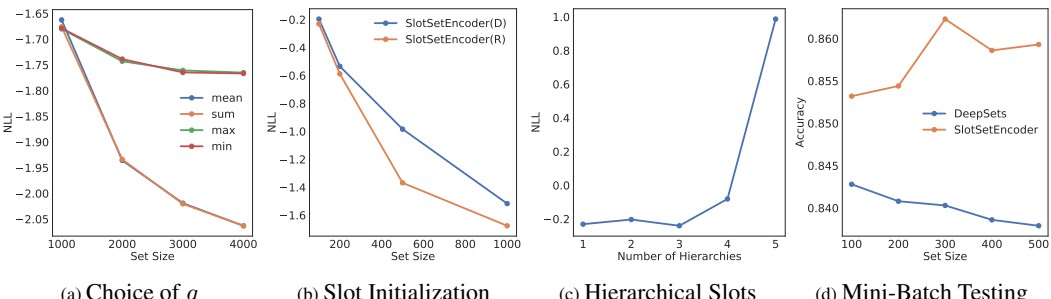

(a) Choice of $g$     (b) Slot Initialization     (c) Hierarchical Slots     (d) Mini-Batch Testing

Figure 4: **(a)** Effects of the aggregation function $g$ on model performance. **(b)** Effects of using random and deterministic slot initialization. **(c)** Effects of stacking multiple hierarchies of the Slot Set Encoder. **(d)** Accuracy of mini-batch testing of MBC models on the ModelNet40 dataset.

more data arrives justifying our initial motivation for making use of all the available set elements instead of using a randomly sampled subset (in the preprocessing stage) in place of the full set.

**Mini-Batch Training of Mini-Batch Consistent Set Encoders** In Section 2.5, we proposed to train the Slot Set Encoder by approximating the full training procedure on partitions of the set. As we highlighted, under memory constraint and large set size assumptions, we cannot take gradient steps with respect to the full set. To demonstrate that this mini-batch training generalizes when the full set is processed at inference time, we train a model on 1000 elements and test it on larger set sizes. This result is demonstrated in Figure 3b on the CelebA dataset where the model performance improves as larger sets are processed. In Figure 3c, we take the same model and evaluate it on smaller set sizes of 100-500 context points and compare it to both DeepSets and Set Transformers where again we find that the Slot Set Encoder perform significantly better. Additionally, Figure 3d shows that the models trained in this fashion can achieve a faster convergence rate. We find that for the problems we consider in this work, this mini-batch training is sufficient and generalizes at inference time.

**Choice of Aggregation Function** In Section 2.3, we stated that we choose the aggregation function $g$ from the set $\{\texttt{mean, sum, max, min}\}$. On the CelebA experiments, we explore the effects of making such a choice for the model with random slot initialization. In Figure 4a, we take a model trained on sets with cardinality 1000 and evaluate it on larger sets at test time. As can be inferred, the aggregation functions $\texttt{mean}$ and $\texttt{sum}$ consistently outperform the $\texttt{min}$ and $\texttt{max}$ for this task. This is intuitive since in the image reconstruction task, an aggregation function that considers the contribution of all pixels is necessary. In Section 3.2, we use a model with $\texttt{max}$ as the aggregation function for Point Cloud Classification where it performs better than the other options. We find that the choice of the aggregation function $g$ is very much informed by the task at hand. We observe a similar trend for the Slot Set Encoder with deterministic slot initialization and report this result in the Appendix.

**Deterministic (D) vs Random (R) Slot Initialization** For slot *initialization*, we have the choice of using deterministic or random (with learnable $\mu$ and $\sigma$) slots. In this choice, we find in almost all our experiments that random initialization performs better across all the tasks we investigated. This trend can be observed in Figure 4b where using random slot initialization results in lower negative log-likelihoods. To explain this behaviour, we reiterate that $slots$ attend to different portions of the input set and hence the usage of random initialization encourages the model to attend to a wider coverage of the input sets during training as opposed to deterministic slot initialization.

**Number of Slots** When we define slots $S \in \mathbb{R}^{K \times h}$, we must decide the number of slots $K$ to use and we explore the effects of this choice on model performance here. We fix a model that encodes an input set to a 64 dimensional vector and train multiple models with varying $K$ (specifically, we test for $K = 1, 32, 64, 128, 256$) and we observe that using very small or large number of slots can have a negative effect on the performance of the model for both deterministic and random slot initialization. For this specific model, we find that selecting $K$ between 32 and 128 performs best. Additionally, we find that this parameter is also very dependent on the task. More details are provided in the Appendix.

**Dimension of Slots** Additionally, for slots $S \in \mathbb{R}^{K \times h}$, we can choose the dimension of each slot $h$ arbitrarily since the projection layer will eventually reduce it to $\hat{h}$. Similarly to the experiments on the number of slots, we fix a model that encodes sets to 64 dimensions using a single Slot Set Encoder and vary the dimensions of the slot keeping $K = 1$. We experiment with $d = 32, 64, 128, 256$ and find that this parameter has a significant impact on the performance of the model. Specifically, we

Table 1: Test Accuracy for Point Cloud Classification with varying set sizes.

| Model | 100 | 200 | 500 | 1000 | 5000 |
|-------|-----|-----|-----|------|------|
| DS | $0.8428 \pm 0.0027$ | $0.8518 \pm 0.0032$ | $0.8574 \pm 0.0017$ | $0.8528 \pm 0.0020$ | $\mathbf{0.8883 \pm 0.0008}$ |
| ST | $0.8258 \pm 0.0042$ | $0.8443 \pm 0.0023$ | $0.8545 \pm 0.0019$ | $0.8511 \pm 0.0018$ | $0.8604 \pm 0.0009$ |
| SSE(D) | $\mathbf{0.8556 \pm 0.0048}$ | $0.8633 \pm 0.0038$ | $\mathbf{0.8710 \pm 0.0040}$ | $0.8720 \pm 0.0009$ | $0.8784 \pm 0.0023$ |
| SSE(R) | $0.8532 \pm 0.0009$ | $\mathbf{0.8667 \pm 0.0028}$ | $0.8699 \pm 0.0020$ | $\mathbf{0.8743 \pm 0.0021}$ | $0.8853 \pm 0.0012$ |

find that for slots with random initialization, the dimension of the slots can have a negative impact if not chosen properly. For deterministic initialization, increasing the slot dimension generally results in better performance. In our experiments involving randomly initialized slots, we train multiple models with varying $h$ and pick the best one. The results discussed here can be found in the Appendix.

**Hierarchical Slot Set Encoder** Finally, we explore stacking multiple layers of Slot Set Encoders based on the hierarchical formulation presented in Section 2.4. We start with a single Slot Set Encoder with $K = 1$ and $h = 32$ and for each hierarchy, we double the slot dimension. For this experiment, we find that stacking multiple Slot Set Encoders *for the tasks we consider* provides marginal performance gains and can result in performance degradation when the hierarchy gets very deep. This is shown in Figure 4c where the performance of the model degrades after 4 hierarchies.

## 3.2 Point Cloud Classification

Using the ModelNet40 dataset, we train models that encode an input set (point cloud) to a single set representation that is used to classify the object as one of 40 classes. We follow the experimental settings of Zaheer et al. [14] and Lee et al. [6] where for point clouds with cardinality less than 5000, all the compared methods encode the point cloud to a 256 dimensional vector and 512 for point clouds with 5000 points or more. Set Transformer and DeepSets use a proprietary version of the ModelNet40 dataset however in our experiments, we use the raw ModelNet40 dataset which can sample an arbitrary number of points, and sample points for each instance randomly at every iteration.

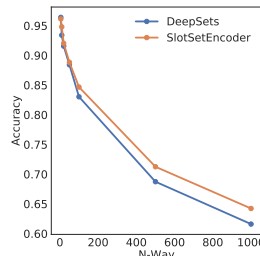

Figure 5: Accuracy vs. 'way' in the centroid prediction task. As the problem becomes harder, our Slot Set Encoder outperforms DeepSets.

The ModelNet40 dataset is difficult to classify when the number of elements in the set is relatively small. This is shown in Table 1 where we test the two variants of our model with both deterministic (D) and random (R) slot initialization. The Slot Set Encoder models outperform DeepSets and Set Transformer. As the number of points increase to 5000, DeepSets and the Slot Set Encoder model achieve similar performance as expected. From the Table 1, it can be seen that the two competing models are DeepSets and our models and hence we check the generalization of both models in the Mini-Batch testing setting where a model trained on sets with 100 elements is tested on 200, 300, 400 and 500 elements. As can be seen from Figure 4d, although DeepSets is Mini-Batch Consistent, it experiences performance degradation when the set size grows on the ModelNet40 dataset. Conversely, the Slot Set Encoder based model gets better as the set size increases at test time showing that it generalizes better than DeepSets with Mini-Batch training (Section 2.5).

## 3.3 Cluster Centroid Prediction

In order to show the effectiveness of Mini-Batch Consistent set encoding on sets of high cardinality, we experiment with Prototypical Networks [11] for classification on the ImageNet and MNIST datasets (see the Appendix for MNIST results). We use a pretrained ResNet50 backbone with the final layer removed, and train the set-encoding layers for DeepSets/SSE in the latent feature space. We train for 20-way classification with 2 support and 2 query instances. Each support set is encoded to a 128 dimensional vector (centroid), and we then train the model to minimize the Euclidean distance between the query instances and the corresponding class centroid. There is no decoder in this experiment, as we only utilize the pooled representation for inference. We investigate the effects of testing on both lower and higher 'way' (5-1000 way) problems (Figure 5) and different encoded support set sizes (Figure 6). Given the large size of the set, we compare our model with DeepSets, as it is the only other MBC set encoder. In Figure 6 we plot accuracy vs. encoded set size for various 'ways'. In both the 1 and 20 way tasks, our SSE and Deepsets show similar performance, we attribute

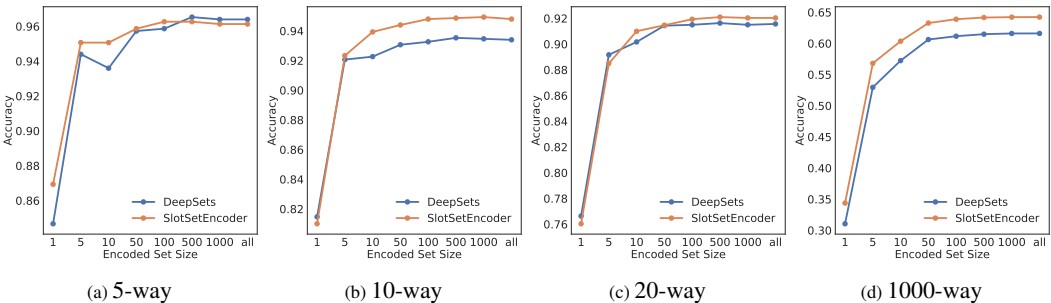

|  | |
|:--:|:--:|
| (a) 5-way | (b) 10-way |
| (c) 20-way | (d) 1000-way |

Figure 6: Accuracy vs. Encoded set sizes for different n-way classification problems on ImageNet. Our Slot Set Encoder performs best on n-way problems which differ from what was seen during training (trained on 20-way). 1-way provides a singleton set, and therefore is about equal for both models.

this to the fact that 1 way classification provides minimally informative singleton sets, and 20-way classification is the setting in which the model was trained. The Slot Set Encoder generalizes better on the other 10 and 1000 way experiments which differ from the training setting.

## 4   Related Work

**Set Encoding**   In DeepSets, Zaheer et al. [14] show that neural architectures for set data are required to obey the permutation invariance/equivariance symmetry and can be formulated as the following sum-decomposition of two functions, $f(\mathbf{x}_n) = \tilde{f}\left( \sum_{i=1}^{n} \phi(\mathbf{x}_i)\right)$ where $\phi$ is a feature extractor and $\tilde{f}$ is some non-linear activation function such as ReLU. Additionally, instead of the sum operator, min, max and mean also satisfy the imposed constraints. While there exist previous works [9] on invariant pooling methods, DeepSets provided a universal model for constructing such neural networks. However, the architectures proposed in DeepSets are simple and in some cases lack expressivity. Set Transformer [6] solves this issue by formulating a set compatible version of the Transformer [12] which is capable of modeling pairwise interactions among the input set. Additionally, Lee et al. [6] reduce the complexity of Transformers from $\mathcal{O}(n^2)$ to $\mathcal{O}(mn)$ by introducing inducing points that serve as queries to reduce the size of the input set before self-attention. Other methods that tackle the set encoding problem include  Murphy et al. [10] and  Zhang et al. [15]. However all of these methods require processing all elements of the input set at once and hence cannot be used in the MBC setting.

**Slot Attention**   Locatello et al. [8] introduced a slot attention mechanism to learn object-centric representations.  Slots are variables which adapt and bind to objects (sets of features) to form non-stationary, input-dependent representations. The slot attention mechanism works by iteratively updating an initialized set of slots with a GRU [1] to adapt them to the current inputs. Through the adaptation process, the slots compete for sets of similar features which compose objects by specializing their weights to achieve a higher activation value in the softmax of the attention matrix. While the idea of slots provided an initial inspiration for our work, they were not designed for set encoding like DeepSets and break MBC for a couple of reasons: 1) The GRU iteratively updates the slots, which depends heavily on the set elements. 2) the Softmax normalization layer depends on all elements in the set and 3) updates on hidden states in the model also break MBC.

## 5   Conclusion

In this work, we have identified a key limitation of set encoding methods when applied to large scale / streaming set encoding. Additionally, we identified a key property, Mini-Batch Consistency, that is required to guarantee that set encoding methods are amenable to mini-batch processing of set. We further detailed the formulation of an instance of an attentive set encoding mechanism dubbed Slot Set Encoder that respects the symmetries of permutation invariance/equivariance, is Mini-Batch Consistent and can be trained with a mini-batch approximation of the full set gradients. We demonstrated the utility of SSE with extensive experiments and ablation studies and verified that

SSE is capable of learning rich set representations while satisfying MBC. An interesting direction of future research would be to use more sophisticated methods to approximate the mini-batch training routine (in Section 2.5) with respect to the full set without having to consider all elements in the set.

# 6 Acknowledgment

This work was supported by the Engineering Research Center Program through the National Research Foundation of Korea (NRF) funded by the Korean Government MSIT (NRF-2018R1A5A1059921), the Institute of Information & communications Technology Planning & Evaluation (IITP) grant funded by the Korea government (MSIT) (No.2020-0-00153), the National Research Foundation of Korea (NRF) funded by the Ministry of Education (NRF-2021R1F1A1061655), the Institute of Information & communications Technology Planning & Evaluation (IITP) grant funded by the Korea government(MSIT) (No.2021-0-02068, Artificial Intelligence Innovation Hub), the Institute of Information & communications Technology Planning & Evaluation (IITP) grant funded by the Korea government(MSIT) (No.2019-0-00075, Artificial Intelligence Graduate School Program(KAIST)) and the HPC Support Project, supported by the Ministry of Science and ICT and NIPA.

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
