# Appendix: Mini-Batch Consistent Slot Set Encoder for Scalable Set Encoding

**Andreis Bruno**[1], **Jeffrey Ryan Willette**[1], **Juho Lee**[1,2], **Sung Ju Hwang**[1,2]

KAIST [1], South Korea
AITRICS [2], South Korea
{andries, jwillette, juholee, sjhwang82}@kaist.ac.kr

## A   Organization

The supplementary file is organized as follows: First in Section B, we provide a more detailed version of Algorithm 1 in the main paper. In Section C, we provide proofs for Proposition 1 and 2 and more details about the mini-batch consistency of baseline models. Section D is on the Image Reconstruction experiments with the CelebA dataset where we provide details of the model architectures used. Additionally, we reproduce the plots in the main paper and include the deterministic slot initialization versions on the model together with the plots for Slot Dimension and Slot Size mentioned in the main text. Section E provides the reference for the model architectures used for the Point Cloud classidication experiments with the ModelNet40 dataset. Finally in Section F, we provide further details on the Cluster Centroid prediction problem.

Unless otherwise specified, all models were trained with the Adam optimizer with learning rate of $1e-3$ and weight-decay of $5e-4$. Additionally, all models are trained for 200 epochs with MultiStep learning rate scheduler at $0.2, 0.6, 0.8$ milestones of the full 200 epochs.

### A.1   Limitations & Societal Impact

**Limitations.**   As stated in section 2.5, our work considers a training time gradient which is an approximation to the gradient of a larger set which must be processed in batches. While effective, there could be opportunities for future research to approximate the gradient of the full set through episodic training, or other strategies which could improve performance.

**Societal Impact.**   Expressive set functions and have the potential for improving fairness in AI by creating more informative contributions from set elements. A permutation equivariant, sum-decomposable function such as Deep Sets grants equal weights to each latent feature, while attention used by both [3] and SSE could provide ways for underrepresented set elements to have more impact on the final prediction. Other than generalization error risk shared by all predictive models, we are unaware of any potential negative societal impacts of our work.

## B   Slot Set Encoder

Below, we provide a version of the Slot Set Encoder Algorithm with finer detail than space allowed in the main text.

35th Conference on Neural Information Processing Systems (NeurIPS 2021).

**Algorithm 1** Slot Set Encoder. $X = \{X_1, X_2, \ldots, X_p\}$ is the input set partitioned into $p$ chunks. $S \in \mathbb{R}^{K \times h}$ are the initialized slots and $g$ is the choice of aggregation function.

---

1: **Input:** $X = \{X_1, X_2, \ldots, X_p\}, S \in \mathbb{R}^{K \times h}, g$
2: **Output:** $\hat{S} \in \mathbb{R}^{K \times d}$
3: **Initialize** $\hat{S}$
4: $S = \texttt{LayerNorm}(S)$
5: $q = \texttt{Linear}_q(S)$
6: **for** $i = 1$ **to** $p$ **do**
7: $\quad k = \texttt{Linear}_k(X_i)$
8: $\quad v = \texttt{Linear}_v(X_i)$
9: $\quad M = \frac{1}{\sqrt{\hat{d}}} * k \cdot q^T$
10: $\quad \texttt{attn} = \texttt{Sigmoid}(M) + 1e - 8$
11: $\quad W = \texttt{attn}/\texttt{attn.sum(dim=2)}$
12: $\quad \hat{S}_i = W^T \cdot v$
13: $\quad \hat{S} = g(\hat{S}, \hat{S}_i)$
14: **end for**
15: **return** $\hat{S}$

---

## C  Mini-Batch Consistency

We restate Propositions 1 & 2 here.

**Property 1** (Mini-Batch Consistency). *Let $X \in \mathbb{R}^{n \times d}$ be partitioned such that $X = X_1 \cup X_2 \cup \ldots \cup X_p$ and $f : \mathbb{R}^{n_i \times d} \mapsto \mathbb{R}^{d'}$ be a set encoding function such that $f(X) = Z$. Given an aggregation function $g : \{Z_j \in \mathbb{R}^{d'}\}_{j=1}^p \mapsto \mathbb{R}^{d'}$, $g$ and $f$ are Mini-Batch Consistent if and only if*

$$g\big(f(X_1), \ldots, f(X_p)\big) = f(X)$$

**Proposition 1.** *For a given input set $X \in \mathbb{R}^{n \times d}$ and slot initialization $S \in \mathbb{R}^{K \times d}$, the functions $f$ and $g$ as defined in Algorithm 1 are Mini-Batch Consistent for any partition of $X$ and hence satisfy Property 1.*

**Proposition 2.** *Let $X \in \mathbb{R}^{n \times d}$ and $S \in \mathbb{R}^{K \times d}$ be an input set and slot initialization respectively. Additionally, let $\texttt{SSEncoder}(X, S)$ be the output of Algorithm 1, and $\pi_X \in \mathbb{R}^{n \times n}$ and $\pi_S \in \mathbb{R}^{K \times K}$ be arbitrary permutation matrices. Then,*

$$\texttt{SSetEncoder}(\pi_X \cdot X, \pi_S \cdot S) = \pi_S \cdot \texttt{SSEncoder}(X, S)$$

### C.1  Proof of Proposition 1

We show that Algorithm 1 satisfies Property 1 by showing that each component of Algorithm 1 satisfies Property 1.

*Proof.* **Linear Layers.** Since all $\texttt{Linear}$ layers in Algorithm 1 are row-wise feed-forward neural network, they all satisfy Property 1 since we can use concatenation to aggregate their outputs for any partition of the inputs. That is, all Linear layers in Algorithm 1 satisfy Property 1.

**Dot Product. Equation 3 & 4** compute the dot product over slots which involves the sum operation. Since the same slots are used for all elements in the input set, the dot product satisfies Property 1 using similar arguments on Linear layers. The sum operation clearly satisfies Property 1.

**Choice of $g$.** Any choice of $g$ from the set $\{\texttt{sum, mean, max, min}\}$ satisfies Property 1.

Since Algorithm 1 is a composition of only the functions above, we conclude that Algorithm 1 satisfies Property 1. $\square$

### C.2  Proof of Proposition 2

The proof is very closely tied to that provided in Appendix D of Locatello et al. [5]. The definitions for permutation invariance and and equivariance are from Appendix D of Locatello et al. [5]. We provide it here for completeness.

**Definition 1** (Permutation Invariance). *A function* $f : \mathbb{R}^{M \times D_1} \to \mathbb{R}^{M \times D_2}$ *is permutation invariant if for any arbitrary permutation matrix* $\pi \in \mathbb{R}^{M \times M}$ *it holds that*

$$f(\pi x) = f(x)$$

**Definition 2** (Permutation Equivariance). *A function* $f : \mathbb{R}^{M \times D_1} \to \mathbb{R}^{M \times D_2}$ *is permutation equivariant if for any arbitrary permutation matrix* $\pi \in \mathbb{R}^{M \times M}$ *it holds that*

$$f(\pi x) = \pi f(x)$$

We provide the proof for Proposition 2 by considering each component of Algorithm 1.

*Proof.* **Equation 3.** In Algorithm 1, we compute the following:

$$\text{attn}_{i,j} := \sigma(M_{i,j}) \tag{1}$$

where $\sigma$ is the Sigmoid function. From the definition of the Sigmoid function, it follows that:

$$
\begin{aligned}
\sigma(\pi_s \cdot \pi_k \cdot M_{i,j}) &= \text{Sigmoid}(\pi_s \cdot \pi_k \cdot M_{i,j}) \\
&= \frac{1}{1 + e^{-(\pi_s \cdot \pi_k \cdot M)_{i,j}}} \\
&= \frac{1}{1 + e^{-M_{\pi_{s(i)}, \pi_{k(j)}}}} \\
&= \sigma(M_{i,j})_{\pi_{s(i)}, \pi_{k(j)}}
\end{aligned}
\tag{2}
$$

That is, the Sigmoid function is permutation equivariant and $\pi_s$ and $\pi_k$ are permutation functions applied to $i$ and $j$ respectively.

**Linear Layers.** `Linear` layers are independently applied to each of the inputs and the slots and hence are permutation equivariant.

**Dot Product.** The dot product involves a sum over the feature dimension and hence also permutation equivariant. This makes Equation 3 & 4, which make use of the dot product all permutation equivariant.

**Choice of $g$.** Any choice of $g$ from the set {`sum`, `mean`, `max`, `min`} is permutation invariant.

By combining all these operations, the Slot Set Encoder in Algorithm 1 is permutation invariant with respect to the inputs and equivariant with with respect to a given set of slot initialization. $\qquad\square$

## C.3 Mini-Batch Consistency of Baseline Models

We provide an analysis of previously published set encoding models and whether or not their original formulation satisfies MBC.

| Model | Pass/Fail | Reason |
|---|---|---|
| Deepsets (original) | Fails | Message Passing Layers |
| Set Transformer | Fails | Self-Attention |
| FSPool | Fails | Sorting |

Table 1: Pass/Fail test for Baselines satisfying Property 1

**Message Passing Layers.** The original formulation of Deepsets [8] utilizes simple message passing layers which use a set dependent normalization, where the considered set is only that which is present in the current batch. Specifically, each set in the current batch is normalized by $\phi(x) = f(x) - g(f(x))$, where $g \in \{max, mean\}$. Normalization based on batch statistics creates a dependency on the current batch and therefore causes the model to fail MBC. As it is trivial to make Deepsets satisfy MBC, our MBC version of Deepsets replaced these layers with traditional neural network layers, which satisfy MBC.

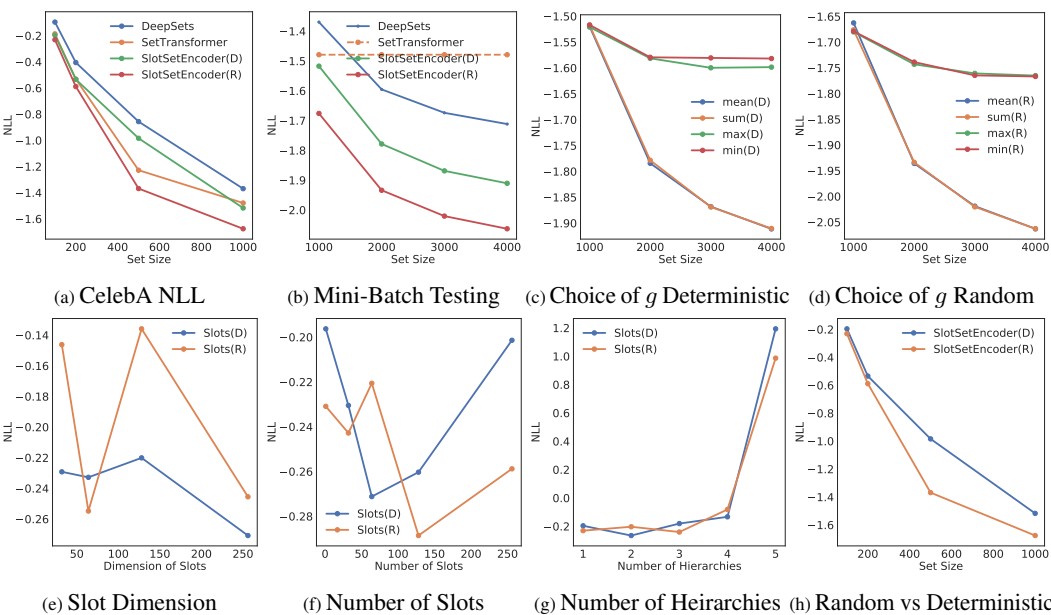

Figure 1: We provide the results in the main paper(Figures 2-6). Additionally, we add the result of the model with deterministic slot initialization.

**Self-Attention.** Self attention used in the Set Transformer [3] allows for modelling complex interactions between set elements, but as depicted in the concept illustration in the main text, creates a dependency on the current batch while constructing the attention matrix $attn(X, X) = \sigma(q(X) * k(X)^\top)$, where $\sigma$ is the softmax function. Both the inner query-key multiplication and the softmax function are dependent on the current batch, thereby failing to satisfy MBC.

# D   Image Reconstruction

In our image reconstruction experiments, we use an encoder-decoder architecture with the encoder followed by a set encoding function. We provide details of the full model in Tables 2 and 3. Additionally, we provide a the plots for the image reconstruction task with the version of the Slot Set Encoder that utilizes deterministic slot initialization together with the plots for the Slot Dimension and Slot Number experiments referenced in the Ablation section in the main paper in Figure 1.

| Layers |
| --- |
| Linear(in_features=5, out_features=64) $\rightarrow$ ReLU |
| Linear(in_features=64, out_features=64) $\rightarrow$ ReLU |
| Linear(in_features=64, out_features=64) |
| SlotSetEncoder(K=1, h=64, d=64, d_hat=64, g='sum', _slots=Random) |

Table 2: Encoder for Image Reconstruction.

| Layers |
| --- |
| Linear(in_features=128, out_features=128) $\rightarrow$ ReLU |
| Linear(in_features=128, out_features=128) $\rightarrow$ ReLU |
| Linear(in_features=128, out_features=128) $\rightarrow$ ReLU |
| Linear(in_features=128, out_features=3) |

Table 3: Decoder for Image Reconstruction

# E   Point Cloud Classification

For the ModelNet experiments on point cloud classification, we use the same architecture as Zaheer et al. [8] and replace the pooling layer with the Slot Set Encoder.

# F   Cluster Centroid Architecture and Extra Results

Below, we provide the exact architecture for the cluster centroid prediction task in section 3.3 for both the Deepsets model and our Mini-Batch Consistent Slot Set Encoder on MNIST/ImageNet. Both models were trained for a total of 50 epochs.

## F.1 MNIST

Each class in MNIST contains roughly 6000 training instances. During training, we randomly sample a set of 5 classes with a support and query set, each consisting 10 instances from each class. Class-wise support sets are encoded to a 128 dimensional vector (centroid). We follow the same training procedure as explained in section 3.3. Exact architectural dtails for the MNIST experiments can be found in Tables 4 and 5. For evaluation, we investigate the effects of encoding different set sizes (1-6000) on model performance. In Figure 3 we show the effect of the encoded set size on the centroid location and observe that the centroid becomes more accurate as the set size increases. Precise centroids also lead to higher classification accuracy, highlighting the utility of encoding sets with high cardinality. This is further demonstrated in Figure 2 where one can see the Slot Set Encoder performs better on all set sizes compared to DeepSets.

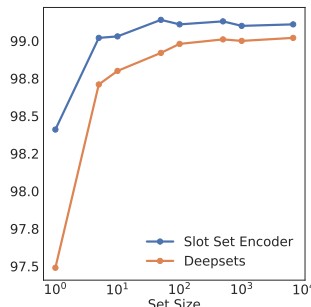

Figure 2: Accuracy vs. set size in the centroid prediction task.

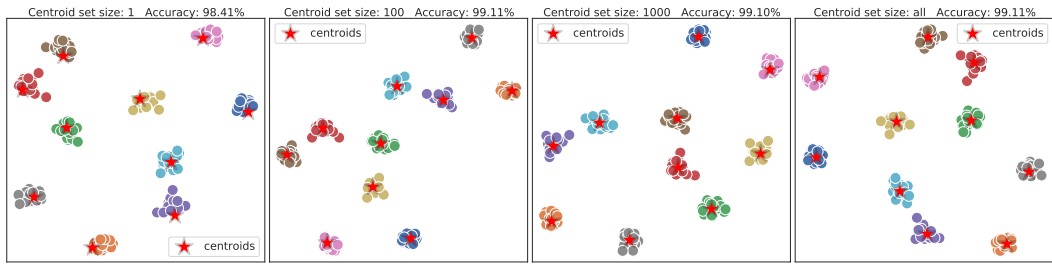

Figure 3: TSNE embeddings of centroids (stars) and classes (circles) in prototypical network [6] classification. (left) a single instance, and (right) the entire training set for each class is encoded into a class centroid which is then used to classify the test set instances via Euclidian distance. As the set size used to predict the centroid increases, centroid location and classification accuracy both rise.

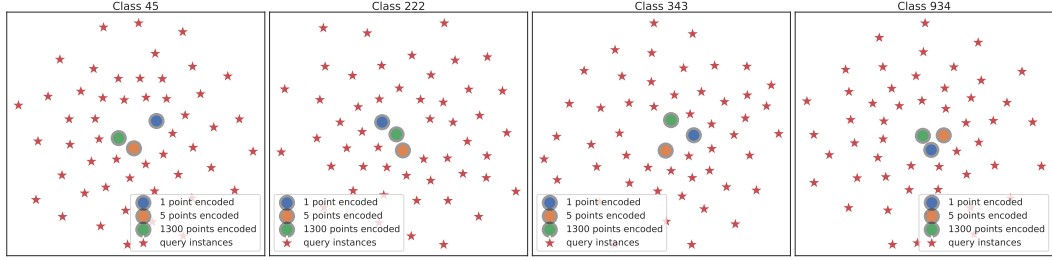

Figure 4: TSNE embeddings of centroids and query set in prototypical network [6] classification. As the number of query instances embedded goes up, the centroid becomes more accurate. (See Figures 5 and 6 in the main text)

| Layers |
| --- |
| Conv2d(1, 32) $\to$ BatchNorm $\to$ LeakyReLU $\to$ MaxPool(2) |
| Conv2d(32, 64) $\to$ BatchNorm $\to$ LeakyReLU $\to$ MaxPool(2) |
| Conv2d(64, 128) $\to$ BatchNorm $\to$ LeakyReLU $\to$ AvgPool |
| SlotSetEncoder(K=32, dim=128) |
| SlotSetEncoder(K=16, h=128, d=128, $\hat{d}$=128, g='mean') |

Table 4: Architecture used for our Mini-Batch Consistent Slot Set Encoder on the MNIST centroid prediction task.

| Layers |
| --- |
| Conv2d(1, 32) → BatchNorm → LeakyReLU → MaxPool(2) |
| Conv2d(32, 64) → BatchNorm → LeakyReLU → MaxPool(2) |
| Conv2d(64, 128) → BatchNorm → LeakyReLU → AvgPool |
| DeepsetsMeanPooling(dim=setdim) |

Table 5: Architecture used for Deepsets on the MNIST centroid prediction task.

| Layers |
| --- |
| ResNet50(pretrained=True) |
| SlotSetEncoder(K=128, dim=128) |

Table 6: Architecture used for SlotSetEncoder on the ImageNet centroid prediction task.

| Layers |
| --- |
| ResNet50(pretrained=True) |
| Linear(128) → ReLU → Linear(128) |
| DeepsetsMeanPooling(dim=setdim) |

Table 7: Architecture used for Deepsets on the ImageNet centroid prediction task.

## G  Datasets

We evaluate our model on the ImageNet [1], CelebA [4], MNIST [2] and ModelNet40 [7] datasets. MNIST consists of 60,000 training images of the handwritten digits 0-9 along with 10,000 test instances. CelebA consists of more than 200000 face images of celebrities with a wide variety of pose variations and backgrounds. ModelNet40 is as 3D CAD dataset with vertices for rendering common object with 40 classes. Point clouds can be constructed from these vertices by sampling points from the vertices. The dataset consists of 9843 training instances and 2468 test instances.