# OpenReview forum: "Mini-Batch Consistent Slot Set Encoder for Scalable Set Encoding"
_NeurIPS.cc/2021/Conference — NeurIPS 2021 Poster_

### Official Review · Reviewer_U9Cv · 2021-07-09

**Rating:** 8
**Confidence:** 4

**Summary:**

This paper introduces a new concept called “mini-batch consistency” (MBC), which is presented as necessary for efficiently training large-scale set encoding models using mini-batch processing of sets.  MBC is appropriate when the cardinality of the sets in the data is very high, and thus all set elements cannot be loaded into memory, or when the data arrives in a stream.  MBC adheres to the required properties of invariance and equivariance for set encoding models.  A MBC-based set encoding mechanism is also proposed, called the slot set encoder.  Extensive experimental results are provided, which demonstrate that the proposed method is scalable and generally outperforms competing approaches.

**Limitations And Societal Impact:**

The authors have included a short, but adequate, discussion of the limitations of their work in Section 5 and Appendix A.1.  Potential negative societal impacts are adequately discussed in Appendix A.1.

**Main Review:**

The MBC property, along with the proposed slot set encoding mechanisms, appear to be technically sound and novel.  The paper is well written, with a clear presentation of the proposed approaches.  Scalable mini-batch training of set encoding models is a relevant problem, and the contributions in this paper are significant, and others will likely build on these ideas in future work.  The theoretical contributions (Propositions 1 and 2) are sound, and the proofs of both propositions are straightforward and appear to be correct.  The experimental results are convincing, and clearly demonstrate that the proposed slot set encoder approaches outperform competing approaches when trained on mini-batches of sets.

A few additional remarks:
- It would be helpful for the authors to include a comparison of the memory consumption (or space complexity) for their slot set encoder approach vs. competing approaches.  Such an analysis would make the memory consumption advantages of their approach clear, particularly for large datasets where mini-batch training has significant advantages.
- It is somewhat surprising that the hierarchical slot set encoder approach gives only marginal gains in predictive performance at best, as shown in Figure 4c.  Do the authors have an explanation for this behavior?


**Time Spent Reviewing:**

5 hours

---

> ### Author Response · Authors · 2021-08-07
> **Response**
>
> We would like to thank the Reviewer for the thorough feedback and useful suggestions. We provide responses to the queries made below.
>
> **1. It would be helpful for the authors to include a comparison of the memory consumption (or space complexity) for their slot set encoder approach vs. competing approaches. Such an analysis would make the memory consumption advantages of their approach clear, particularly for large datasets where mini-batch training has significant advantages.**
>
> We provide memory consumption requirements for the point cloud experiments using 5000 elements in a set. All memory values are read directly from an Nvidia GeForce GTX 1080p graphics card with 11GB of memory using the nvidia-smi command. We compare between DeepSets, Set Transformer and Slot Set Encoder. To train on a point cloud with 5000 elements, DeepSets requires ~5GB of memory, Set Transformer requires ~14GB of memory and Slot Set Encoder requires ~10GB of GPU memory.
> In the streaming setting where we use mini-batch training with mini-batch sizes of 1000, 500, 200 and 100 the Slot Set Encoder model requires ~5GB of memory to train for all these mini-batches. That is, with mini-batch training, we require less than half the memory required to train on the full set and our memory requirements become comparable to DeepSets while obtaining better performance.
>
> **2. It is somewhat surprising that the hierarchical slot set encoder approach gives only marginal gains in predictive performance at best, as shown in Figure 4c. Do the authors have an explanation for this behavior?**
>
> In the current formulation of the hierarchical slot set encoder, we can stack multiple slot set encoders (as in Figure 4c) but we are unable to apply non-linear transformations to the slot output at each step in the hierarchy as Set Transformer does between multiple PMA or ISAB layer (except for the slot set encoder in the top hierarchy) since this breaks MBC. As a result, as Reviewer s6Je also points out, this behaviour is expected. One way to resolve this issue, as future work, would be to relax MBC to obtain an ‘approximately’ MBC model that can exploit the hierarchical slot set encoder. For instance, we find that by adding a non-linear activation layer such as ReLU after each slot set encoder in the hierarchy we are able to obtain better performance as the hierarchy increases but since this model breaks MBC, we do not present these results in the paper and leave such exploration as future work. We will include these results in the appendix of the final version since it can serve as a starting point for further exploration into ‘approximately’ MBC set encoders. However in the experiments on the cluster centroid prediction problem we use a model with T=2 and obtain good performance (we provided this detail in Table 4 of the appendix. We will make this more explicit in the updated version of the paper). Finally, while the hierarchical SSE model is interesting, we believe (and as Reviewer DaP6 also iterates) that this work can be a first step towards a new class of algorithms that relax the MBC property and can take full advantage of the hierarchical structure presented in Section 2.4.

---

> > ### Comment · Reviewer_U9Cv · 2021-08-18
> > **Rebuttal response**
> >
> > Thank you for the response to my questions/comments.  I’ve read through the other reviews and rebuttal comments, and am satisfied with the remarks.  I think the authors have adequately addressed the questions and issues raised by the other reviewers, especially reviewers P7K7, s6Je, and 9cx8.  In particular, the response to reviewer 9cx8, who is the most critical, is quite thorough and appears to largely address this reviewer’s concerns.  Therefore, I maintain my review score of 8 (Top 50% of accepted NeurIPS papers, clear accept).

---

> > > ### Author Response · Authors · 2021-08-18
> > > **Thank you for your time and effort**
> > >
> > > We thank you for taking the time to carefully review our work and evaluate our responses to all reviewers.

---

### Official Review · Reviewer_9cx8 · 2021-07-12

**Rating:** 6
**Confidence:** 3

**Summary:**

The paper addresses the problem of set encoding when the set is too large to fit in the memory, or given as a stream of data. The proposed method randomly partitions the set,  encodes each random subset and then aggregates all the encodings. The key contribution of the proposed method is that the aggregation of the encodings is required to be equal to the encoding of the full set and it satisfies permutation invariance and equivariance. This property is formally defined and named as Mini-Batch Consistent Set Encoding in the paper. Compared to DeepSets, which also satisfy Mini-Batch Consistent Set Encoding, the proposed method can model pairwise interactions, while DeepSets lack this property. The encoding is implemented using previously introduced slot attention [8] (the attention is done over “slots” instead of self-attention) with the two differences: softmax is replaced with sigmoid in constructing the attention matrix and GRU update of slots is omitted here. The experiments compare the proposed method with two previous methods for set encoding in three tasks showing some improvement.

**Limitations And Societal Impact:**

yes

**Main Review:**

I have two main problems with the paper. One is the technical novelty of this paper, which in my opinion is very modest which is extending previous methods for set encoding by parts with the slot attention (utilized from another previous work). The  Hierarchical Slot Set Encoder that is the most novel part of the paper (in my opinion) doesn’t seem to show much improvement.
My second problem is with the clarity of the paper. First, it assumes prior knowledge about slots. It would be very helpful to give some intuitive explanation about the mechanism that is central to this paper. Second, I was confused about the use of word “partitions”. First, I thought that it creates multiple partitions of the set as a part of the algorithm, but then I realized that the authors use the word “partitions” to refer to sub-sets of the same partition.
Also I cannot agree with the claim (in the contributions part of the paper) that experiments are extensive. All the experiments are contrived, if there is a real need in the proposed method, it would be more convincing to include an experiment in which the conditions assumed in the paper are satisfied. The motivation was the data from the LHS, but the experiments show only standard vision applications.

**Time Spent Reviewing:**

2.5

---

> ### Author Response · Authors · 2021-08-07
> **Response**
>
> We would like to thank the Reviewer for the review. We find that there has been some critical misunderstanding of our work and we provide detailed responses to all the issues raised below.
>
> ---
>
> **1. The technical novelty is very modest**
>
> This seems like a misunderstanding coming from **overlooking the main novelty**of our work, which is the proposal of a **large-scale set encoder**and the **properties that should be met for proper set encoding**under such a setting. This novelty is well-acknowledged by other reviewers (Reviewer P7K7 and U9Cv states that our work is novel, and the latter mentions that it will likely to be further studied by others in the future. Reviewer DaP6 commented that this setting has thus far not been considered in the set representation learning community). We briefly recap the novelty of our work in the multiple aspects below:
>
>  - We tackle the problem of large-scale set encoding for the first time, which is a practical real-world problem but has been neglected by previous works on set encoding. Existing works such as DeetSets and Set Transformer simply **bypassed this scalability issue**by experimenting with smaller sets that can fit into  memory.
>
>  - We identify and formalize a key property, **Mini-Batch Consistency (MBC)**, required for provable consistent large scale set encoding. A number of challenges arise when designing a set encoder that satisfies MBC such as maintaining permutation invariance and equivariance between minibatches. We then empirically show that set encoders such as Set Transformer that violate MBC perform poorly in the large scale and streaming setting (Figure 2).
>
>  - Finally, we design and implement a working mini-batch consistent set encoder, the **Slot Set Encoder**. We want to emphasize that our Slot Set Encoder is **not the same**as Slot Attention [1], and that the concept of slots even precedes Slot Attention (see [2]). Slot Attention is not a set encoding method, and a straightforward application of it **will not satisfy the mini-batch consistency**, due to the use of softmax function and the interaction across the slots with the iterative GRU updates. Thus, it is nontrivial to design and implement the Slot Set Encoding mechanism to satisfy mini-batch consistency, and saying that Slot Set Encoder is a simple modification of Slot Attention, is like saying that Set Transformer is a simple modification of a Transformer.  We further discuss the differences between our Slot Set Encoder and Slot Attention in the remaining responses.
>
> ---
>
> **2. ‘This work extends previous methods for set encoding by parts with the slot attention (utilized from another previous work)’**
>
> While we utilize slots in SSE, the resulting algorithm is very different from the Slot Attention. Slot Attention [1] was developed for representation learning and not set encoding. Moreover, Slot Attention is not mini-batch consistent, and SSE does not attempt to use it for set encoding. In Section 4 (lines 355-364) we explicitly state that while the idea of slots provided the initial inspiration for our work, Slot Attention (which is an application of slots itself) was not designed for set encoding like DeepSets and breaks mini-batch consistency for the following reasons: 1) The GRU layer iteratively updates the slots and heavily depends on the set elements. One has to have the full set available in order to process them with Slot Attention. In the resource constrained and streaming setting, this is not desirable. 2) the Softmax normalization layer depends on all elements in the set to compute the statistics and 3) updates to the hidden states in Slot Attention again rely on the full set and break MBC. In summary, while our work and Slot Attention both utilize slots, they are two different algorithms designed for two different purposes. In this regard, we do not agree that the usage of slots affects the novelty of the method we present. Slots are just a tool for realizing an MBC set encoder.
>
> ---
>
> **3. I was confused about the use of word “partitions”. First, I thought that it creates multiple partitions of the set as a part of the algorithm, but then I realized that the authors use the word “partitions” to refer to subsets of the same partition.**
>
> This is factually incorrect and is a **critical misunderstanding** of what is being done. When we use the word ‘partitions’ **we do indeed refer to partitions of the set**and do not refer to the subsets of the same partition. In Section 2.1 (lines 93-95) we explicitly state that we randomly partition a given set to obtain proper subsets. Since sets are partitioned into sections that are processable (from a computation or resource constraint perspective), there is no need to re-partition the subsets. In Section 2.5, we present an approximate Mini-batch training routine for MBC set encoders which approximates gradients with random partitions but even here, we do not repartition these random partitions. We will further clarify this in the revision.
>
> ---
>
> **4. First, it assumes prior knowledge about slots. It would be very helpful to give some intuitive explanation about the mechanism that is central to this paper.**
>
> We do not assume prior knowledge about slots, and **specifically define what slots are in Section 2.3 (Lines 119-125)**. However, we will include a more intuitive explanation of slots in the approach section, for better understanding.
>
> ---
>
> **5. I cannot agree with the claim (in the contributions part of the paper) that experiments are extensive. All the experiments are contrived, if there is a real need in the proposed method, it would be more convincing to include an experiment in which the conditions assumed in the paper are satisfied.**
>
> Firstly, for the class of problems we consider, we believe that we have provided extensive experiments and analysis of our presented method. We evaluate our method on 4 datasets (MNIST, CelebA, ImageNet, and ModelNet40) and tackle classification, regression and cluster centroid prediction problems and compare our method with the relevant baselines. Reviewers s6Je and U9Cv all comment on the **extensiveness of the experiments as well as the extensive ablation studies**that we perform.
>
> Further, our experiments are not contrived. All the experiments we have presented are standard benchmarks for the tasks and problem domains we consider, and we do not modify the dataset in any ways. Also, for vision tasks, there does exist a need to encode a large set that does not fit into the memory. For instance, in the centroid cluster prediction experiments where we encode an entire dataset, it is impossible to run Set Transformer on a standard GPU with 12GB of memory. The same issue arises with the image reconstruction and point cloud classification tasks. Previous methods simply bypass these real-world problems by simply sampling a subset of the original set, and experimenting with them. Thus, what are contrived are the **experimental settings in previous works**, and our experiments are highly realistic and are more true to real-world settings. Additionally, in the real-world setting, **data arrives in a stream**and our SSE provides a principled way to encode such set-structured data **given as a stream** (see the experiments and analysis in Section 3.1).
>
> ---
>
> **6. The motivation was the data from the LHC, but the experiments show only standard vision applications.**
>
> LHC mentioned in the introduction is only an example, and we did not claim that our work is solely motivated by the encoding of LHC data. Many problems in modern deep learning can be cast into the problem of large-scale set encoding (e.g. point cloud classification, dataset encoding etc.) and we are specifically interested in how to consistently encode such a large set under resource constraints, with provable guarantees.
>
> We provide additional experimental results on a large scale particle physics problem using the Higgs Boson detection dataset [4]. We utilized a deterministic version of [3] and train the model by encoding context sets and decoding them into the weights of a linear model for prediction on the target sets. During testing, we encode the entire training set which consists of 10,500,000 instances and test on 500,000 held out target points. We train for 32,000 episodes, using SGD with momentum. We cannot use Set Transformer in this experiment since it has a large memory footprint. We provide the results below:
>
>
> | Model    | Accuracy | AUROC |
> |----------|----------|-------|
> | DeepSets | 58.1     | .61   |
> | SSE      | 65.2     | .70   |
>
> As can be seen, SSE outperforms DeepSets (the only other MBC model) on this non-vision application and is applicable to encoding of very large sets.
>
> ---
>
> **References**
>
> [1] Locatello, F., Weissenborn, D., Unterthiner, T., Mahendran, A., Heigold, G., Uszkoreit, J., ... & Kipf, T. (2020). Object-centric learning with slot attention. arXiv preprint arXiv:2006.15055.
>
> [2] RIMs, https://arxiv.org/abs/1909.10893
>
> [3] Gordon, J., Bronskill, J., Bauer, M., Nowozin, S., & Turner, R. E. (2018). Meta-learning probabilistic inference for prediction. arXiv preprint arXiv:1805.09921.
>
> [4] https://archive.ics.uci.edu/ml/datasets/HIGGS

---

> > ### Comment · Reviewer_9cx8 · 2021-08-11
> > **Updated review**
> >
> >  I increase my ranking. However,  I still don't see a major contribution in this paper.

---

> > > ### Author Response · Authors · 2021-08-11
> > > **Thank you for your review**
> > >
> > > Thank you for taking the time to re-evaluate our work. We would like to re-iterate that the main contribution of our work is 1) the formalization of the MBC property, which is violated by every set encoding method we are aware of (outside of DeepSets), and 2) The proposal of a new attentive set encoding architecture that is performant and provably MBC. We thank you again for your time and effort in evaluating our work and our responses.

---

### Official Review · Reviewer_DaP6 · 2021-07-16

**Rating:** 7
**Confidence:** 5

**Summary:**

This paper studies the problem of set encoding in the setting where the cardinality of the set is prohibitively large or unbounded. For example, this is the case when dealing with streaming data. A property called Mini-Batch Consistency (MBC) is defined, and it is argued that set encoding algorithms must have the MBC property to learn useful representations of the entire set when partitions of the set are being processed incrementally. A slot-based algorithm that is MBC is proposed. It models interactions between the inputs and slots to learn strong representations. The algorithm, called the Slot Set Encoder (SSE), is compared against Deep Sets and the Set Transformer on a variety of relevant tasks.

**Limitations And Societal Impact:**

The key limitation, which is the inability to model pairwise interactions between set elements, is not mentioned in the Conclusion section. This should be remedied. As mentioned above, I believe this represents a promising direction for an extension to this work.

**Main Review:**

Originality
--------------
- This task has yet to be considered in the set representation learning community.
- The method, SSE, offers a unique capability of processing partitions of sets incrementally while also learning strong representations with input-to-slot attention.
- However, the Introduction and Related Work noticeably are missing some recent papers on set encoding, for example, [1-3], that would help frame the proposed SSE better within the wider context of set encoding methods. These should be discussed in a revised version.


Quality
--------------
- I believe the MBC property, the SSE algorithm, and Propositions 1 and 2 are for the most part technically sound, and the experimental results verify the authors’ claims.
- The output of Algorithm 1 as described is $\hat{S}$, which is a *set* with cardinality $K$. However, as stated explicitly in L160 “The Slot Set Encoder can encode any given set $X \in \mathbb{R}^{n \times d}$ to a $\hat{d}$ dimensional vector representation”. And, when checking the provided code, I saw that $\hat{S}$ is initialized to zeros and additively updated, and then $g \in \{ \texttt{mean}, \texttt{sum}, \texttt{max}, \texttt{min} \}$ is applied to the final $\hat{S}$ to reduce the slot dimension $K$ to 1, so that the output of Algorithm 1 is a vector. The Algorithm 1 description in the paper should be fixed to match the code.
- The paper does a good job of explaining the distinction between the pairwise interactions modeled by Set Transformer’s attention and the input-to-slot attention that SSE learns. I would have liked to see visualizations of what that attention (the $W_{i,j}$) learns in the experiments section.
- Since the inability to model pairwise interactions between set elements could potentially affect the usefulness of this technique, I would have liked to see this highlighted more prominently in the limitations in the Conclusion.


Clarity
--------
- The paper is easy to read and follow.
- Error bars are missing from the line plots and should be added to show robustness across multiple training runs.


Significance
----------------
- I think this work represents a good first attempt at solving the problem of scalable set encoding.
- It does not provide a complete solution to the problem, since the modeling of pairwise interactions between set elements is essential for most domains where set encoding is the prevalent technique, such as in point cloud processing.
- I think this is fine, though, as this raises interesting questions such as whether there are ways to build set encoding algorithms that are guaranteed to be "approximately" MBC and can model "some amount" of pairwise interactions between set elements.


References
---------------
1. Murphy, Ryan L., et al. "Janossy pooling: Learning deep permutation-invariant functions for variable-size inputs." arXiv preprint arXiv:1811.01900 (2018).
2. Zhang, Yan, Jonathon Hare, and Adam Prügel-Bennett. "FSPool: Learning set representations with featurewise sort pooling." arXiv preprint arXiv:1906.02795 (2019).
3. Huang, Qian, et al. "Better set representations for relational reasoning." Advances in Neural Information Processing Systems 33 (2020).

UPDATE POST-REBUTTAL: As explained in my response to the author's rebuttal, my comments have been addressed and therefore I am happy to maintain my initial score of 7.


**Time Spent Reviewing:**

6

---

> ### Author Response · Authors · 2021-08-07
> **Response**
>
> We would like to thank the Reviewer for providing us with detailed feedback and suggestions. We provide answers to the questions raised below.
>
> **1. References to Relevant Literature**
>
> We would like to thank the Reviewer for the suggested related work. We will discuss these in the wider context of set encoding methods in the updated version of the paper and cast them in the context of Mini-Batch Consistent set encoding methods.
>
> When considered in terms of set encoding functions that are MBC:
>
> - Janossey [1] pooling solves the problem of permutation invariance by simply considering all permutations in the pooling function. By definition, all permutations cannot be realized in the MBC setting, therefore Janossey pooling fails MBC.
>
> - FSPool [2] does not satisfy MBC due to the batch dependent sorting of elements. Different partitions of the same set will result in different sorted feature vectors and therefore always result in a different set encoding.
>
> - In general, the approach proposed in [3] would be orthogonal to most set encoding frameworks, but when analyzed under the constraints of MBC, the iterative procedure in Algorithm 1 of [3] would fail to satisfy MBC without having access to the total pooled representation.
>
> We will discuss these methods in the final version of the paper since this will provide a wider coverage of current set encoding methods to the reader as well as highlight the importance of the MBC constraint for large scale set encoding. Thank you for pointing these works out.
>
> **2. The output of Algorithm 1 is a vector**
>
> Indeed the output of Algorithm 1 is a vector only in the case of K=1. However when K>1, the output is a set of vectors. In Algorithm 1, we provide the general slot set encoder and the output is dependent on the choice of K. Thank you for pointing this out, we will make this explicit in the updated version to avoid any confusion.
>
> **3. Since the inability to model pairwise interactions between set elements could potentially affect the usefulness of this technique, I would have liked to see this highlighted more prominently in the limitations in the Conclusion.**
>
> The reviewer is correct in that this is a limitation of SSE since modeling pairwise interactions between set elements directly would violate MBC. We will highlight this in the limitations of the updated version of the paper. However we would like to draw attention to the fact that SSE still outperforms Set Transformer (which does model all pairwise interactions between set elements directly) in Figure 3(a) and Table 1 and hence modeling pairwise interactions over slots may not necessarily be a disadvantage to SSE. We will add this discussion in the conclusion since it can jumpstart further research into approximately MBC set encoding methods.
>
> **4. Suggestions to include visualizations and error bars on line plots.**
>
> We will include visualizations of the slot attention in the updated version of the paper. Also, all values in the plots are over 5 runs each with random initializations. The variances are small and barely visible on the plots. We will include these in the updated version of the paper as suggested for completeness.
>
> **References**
>
> [1] Murphy, Ryan L., et al. "Janossy pooling: Learning deep permutation-invariant functions for variable-size inputs." arXiv preprint arXiv:1811.01900 (2018).
>
> [2] Zhang, Yan, Jonathon Hare, and Adam Prügel-Bennett. "FSPool: Learning set representations with featurewise sort pooling." arXiv preprint arXiv:1906.02795 (2019).
>
> [3] Huang, Qian, et al. "Better set representations for relational reasoning." Advances in Neural Information Processing Systems 33 (2020).

---

> > ### Comment · Reviewer_DaP6 · 2021-08-29
> > **Rebuttal response**
> >
> > The authors have adequately responded to my (~minor) concerns and promised to address them in the final version. After reading the other reviews and responses, I did not see any new criticisms that would cause me to lower my initial score of 7 (Good paper: accept).

---

### Official Review · Reviewer_s6Je · 2021-07-18

**Rating:** 6
**Confidence:** 5

**Summary:**

The authors proposes the use of attention based set encoder which can scale to arbitrary set sizes. Attention in Transformers has the quadratic dependence on number of positions (i.e. cardinality of the set). The proposed method makes the problem easier by enabling the use of slots while preserving a property which method defines as mini-batch consistency. The proposed method is invariant to the order of the subsets, and invariant to permutations on the set elements.



**Ethical Concerns:**

No ethical concerns.

**Limitations And Societal Impact:**

Limitations adequately addressed.

**Main Review:**

Proposed method: The paper proposes the  Slot Set Encoder that respects the symmetries of permutation invariance / equivariance. The basic idea is to represent K slots (can be randomly initialized or the initial representation can be learned). Now, the paper proposes to use the cross-attention between the resulting K slots and over partitions of the input set X

Experiments: The paper evaluates the proposed model model on various datasets  (MNIST, CeleA, ImageNet,  ModelNet40) datasets. The paper compares to various baselines like DeepSets and  Set Transformer, and achieves superior results as compared to both the methods.

Pros:
- The paper is very well written.
- The authors compared to the relevant baselines. The reviewer particularly liked the ablations done in section 3.1.1 that evaluates the proposed model by restricting the memory, showing the importance of aggregating function, and the effect of using random or learned initializations of the slots, effect of the number and dimensionality of the slots.
-  Its interesting that the paper also explores the use of hierarchical set encoding, and shows the result that the performance decreases as we increase the number of layers which intuitively makes sense.

Cons:
- Some of the most relevant work is missing in the current version of the paper. Their are different papers [1, 2] which uses a similar idea (but for different problems). The basic idea is same: learn a encoding function for the set of slots to reduce the computational complexity.  [1] tries it in the context of transformers/modular RNNs. The representation of different positions/RNNs act as a set containing different tensors. The idea is to learn a latent bottleneck consisting of a set of slots. Then there's cross-attention between these set of slots and the original representation of different positions. Thus reducing the complexity of attention from quadratic to linear in the number of positions.

- The idea of cross-attention in the way used in the paper is first used in RIMs [3]. The contribution of Slot attention was to make it iterative.

Relevant Literature work:
- [1] Coordination Among Neural Modules Through a Shared Global Workspace, https://arxiv.org/abs/2103.01197
- [2] Perceiver: General Perception with Iterative Attention, https://arxiv.org/abs/2103.03206
- [3] RIMs, https://arxiv.org/abs/1909.10893

=========================================================

After rebuttal:

I thank the authors for their response. I will keep my original score. Good results in the hierarchical setup, or just carefully constructed experiments for showing improved results would be an interesting area to explore.




Questions for the authors:
- The reviewer is pretty happy to increase my score, if the relevant work is appropriately referenced.

**Time Spent Reviewing:**

5 hours

---

> ### Author Response · Authors · 2021-08-07
> **Response**
>
> We would like to thank the Reviewer for the valuable feedback and suggestions. We provide our response below.
>
> **1. References to Relevant Literature**
>
> We would like to thank the Reviewer for drawing our attention to these works. We agree with the Reviewer that some of the underlying motivations and ideas are similarly motivated and hence we will discuss these works in the updated version of the paper.
>
> - In [3], recurrent independent mechanisms compete with each other by attending to different portions of a given input. Also these recurrent independent mechanisms can be made to interact sparingly. This idea is fundamental and analogous to the way slots as used in this paper operate.
>
> - In [1], again the idea of slots and cross attention over slots are used extensively for complexity reduction.
>
> - In [2], proposes iterative attention allowing the model to scale to very large inputs.
>
> Our work shares commonality with these ideas and we will discuss and place them in the context of this work in the final version of the paper.
>
> **References**
>
> [1] Coordination Among Neural Modules Through a Shared Global Workspace, https://arxiv.org/abs/2103.01197
>
> [2] Perceiver: General Perception with Iterative Attention, https://arxiv.org/abs/2103.03206
>
> [3] RIMs, https://arxiv.org/abs/1909.10893

---

> > ### Comment · Reviewer_s6Je · 2021-08-25
> > **Thanks for the rebuttal.**
> >
> > I would like to thank the authors for the response to my questions/comments.

---

### Official Review · Reviewer_P7K7 · 2021-07-22

**Rating:** 6
**Confidence:** 3

**Summary:**

This paper addresses the problem of set encoding when the dimensionality and/or cardinality of the set it too large to fit into memory to feed into an encoder or when set elements arrive in a stream. This paper defines a property, mini-batch consistency, for set encoding functions that encode the set in non-overlapping subsets (mini-batches) then aggregate the mini-batch encodings to achieve the set encoding. If obeying mini-batch consistency, the encoder achieves the same set encoding as if the encoding function was applied directly to the entire set. Finally, the paper proposes an encoding function, Slot Set Encoder, to satisfy mini-batch consistency.

**Limitations And Societal Impact:**

Limitations with regards to the motivation for SSE as discussed above are not addressed in the paper, ie are slots and hierarchies actually useful in the proposed method.

**Main Review:**

The problem formulation is interesting and appears to be novel. However, the specific benefits of the proposed MBC method, SSE are under motivated in the empirical evaluation.
As compared to Deep Sets [1] which also satisfies mini-batch consistency, SSE’s claimed improvement is modeling set element interactions by proxy of modeling the interaction between slots. However hit seems the using deeper hierarchies hurts performance or has no effect on performance compared to number of hierarchies T=1. Since the number of slots in the T-th hierarchy is set to K_T=1, then SSE is also using one slot and one hierarchy. Thus, slots and hierarchies proposed in the model don’t get exploited or properly motivated. Indeed most performance measures evaluated in experiments are comparable between SSE and Deep Sets, with the exception of generalization to larger sets at test time (Fig 3b) where SSE significantly outperforms Deep Sets.

[1] Zaheer, Manzil, et al. "Deep Sets." Advances in Neural Information Processing Systems 30 (2017).

**Time Spent Reviewing:**

4

---

> ### Author Response · Authors · 2021-08-10
> **Response**
>
> We would like to thank the Reviewer for the review and finding the problem and solutions that we present interesting and novel. We address the questions raised below.
>
> **1. Are slots and hierarchies actually useful in the proposed method?**
>
> In the formulation of the SSE in Section 2.3, we satisfy MBC by utilizing the fact that slots can be shared across all elements. This makes slots fundamental and necessary to our MBC SSE formulation. Additionally, the SSE by itself, without the hierarchical model proposed in Section 2.4, is a valid set encoding function just like DeepSets and Set Transformer and has practical utility on its own with the added advantage of also being mini-batch consistent. This makes SSE our main realization of a set encoding function that is MBC. The hierarchical SSE model presented in Section 2.4 is a natural extension of SSE and in the comments below, provide more details on this extension.
>
> **2. It seems using deeper hierarchies hurts performance or has no effect on performance compared to hierarchies T=1. Since the number of slots in the T-th hierarchy is set to K_T=1, then SSE is also using one slot and one hierarchy. Thus, slots and hierarchies proposed in the model don’t get exploited or properly motivated.**
>
> In the current formulation of the hierarchical slot set encoder, we can stack multiple slot set encoders (as in Figure 4c) but we are unable to apply non-linear transformations to the slot output at each step in the hierarchy as Set Transformer does between multiple PMA or ISAB layer (except for the slot set encoder in the top hierarchy) since this breaks MBC. As a result, as Reviewer s6Je also points out, this behaviour is expected. One way to resolve this issue, as future work, would be to relax MBC to obtain an ‘approximately’ MBC model that can exploit the hierarchical slot set encoder. For instance, we find that by adding a non-linear activation layer such as ReLU after each slot set encoder in the hierarchy we are able to obtain better performance as the hierarchy increases but since this model breaks MBC, we do not present these results in the paper and leave such exploration as future work. We will include these results in the appendix of the final version since it can serve as a starting point for further exploration into ‘approximately’ MBC set encoders. However in the experiments on the cluster centroid prediction problem we use a model with T=2 and obtain good performance (we provided this detail in Table 4 of the appendix. We will make this more explicit in the updated version of the paper). Finally, while the hierarchical SSE model is interesting, we believe  (and as Reviewer DaP6 also iterates) that this work can be a first step towards a new class of algorithms that relax the MBC property and can take full advantage of the hierarchical structure presented in Section 2.4.
>
> **3. Since the number of slots in the T-th hierarchy is set to K_T=1, then SSE is also using one slot and one hierarchy.**
>
> The number of slots in the T-th hierarchy is set to 1 because we want to obtain a single set representation (a single vector) for the full set. Set Transformer and DeepSets also encode to a single vector which is equivalent to setting K=1 in SSE. Also, the usage of a single hierarchy and setting K=1 here ensures that all competing set encoding models have almost the same number of parameters for fair comparison. However, in the supplementary material, Figure 1f studies the effects of increasing K to values larger than 1. We find that we achieve better performance by increasing K up to a certain point. Finding a good value of K should be chosen dependent on the task and validation performance. Please see lines 282-288 in the main text which discusses this in more detail.

---

### Decision · Program_Chairs · 2021-09-27

**Decision:**

Accept (Poster)

**Comment:**

The reviewers have reached a consensus that the paper merits publication, even though some reviewers still rate the paper as borderline. The author responses have done a good job of addressing the reviewers' concerns. I agree with this assessment, and am recommending that this paper should be accepted.